# Digital electronics in fibres enable fabric-based machine-learning inference

Gabriel Loke[1,2,9], Tural Khudiyev[2,9], Brian Wang[3], Stephanie Fu [3], Syamantak Payra [3], Yorai Shaoul[3], Johnny Fung[4], Ioannis Chatziveroglou[3], Pin-Wen Chou[5], Itamar Chinn[3], Wei Yan [2], Anna Gitelson-Kahn[6], John Joannopoulos[7,8] & Yoel Fink [1,2,4,7 ✉]

Digital devices are the essential building blocks of any modern electronic system. Fibres containing digital devices could enable fabrics with digital system capabilities for applications in physiological monitoring, human-computer interfaces, and on-body machine-learning. Here, a scalable preform-to-fibre approach is used to produce tens of metres of flexible fibre containing hundreds of interspersed, digital temperature sensors and memory devices with a memory density of ~$7.6 \times 10^5$ bits per metre. The entire ensemble of devices are individually addressable and independently operated through a single connection at the fibre edge, overcoming the perennial single-fibre single-device limitation and increasing system reliability. The digital fibre, when incorporated within a shirt, collects and stores body temperature data over multiple days, and enables real-time inference of wearer activity with an accuracy of 96% through a trained neural network with 1650 neuronal connections stored within the fibre. The ability to realise digital devices within a fibre strand which can not only measure and store physiological parameters, but also harbour the neural networks required to infer sensory data, presents intriguing opportunities for worn fabrics that sense, memorise, learn, and infer situational context.

[1] Department of Materials Science and Engineering, Massachusetts Institute of Technology, Cambridge, MA, USA. [2] Research Laboratory of Electronics, Massachusetts Institute of Technology, Cambridge, MA, USA. [3] Department of Electrical Engineering and Computer Science, Massachusetts Institute of Technology, Cambridge, MA, USA. [4] Department of Mechanical Engineering, Massachusetts Institute of Technology, Cambridge, MA, USA. [5] Harrisburg University of Science and Technology, Harrisburg, PA, USA. [6] Textile Department, Rhode Island School of Design, Providence, RI, USA. [7] Institute for Soldier Nanotechnologies, Massachusetts Institute of Technology, Cambridge, MA, USA. [8] Department of Physics, Massachusetts Institute of Technology, Cambridge, MA, USA. [9] These authors contributed equally: Gabriel Loke, Tural Khudiyev. ✉email: yoel@mit.edu

Mobile digital computing systems, also known as wearables, are being increasingly used to collect physiological parameters from the surface of the human body. Adoption has been limited[1] by the need to convince prospective users to carry an additional object, leading to the emergence of only a small number of highly specific form factors. In general terms, a wearable system typically involves a fairly rigid device placed over a small area of contact and particular positions on the body which in turn limit the type of data that these devices can access. Without digressing much into semantics, it is noteworthy that the term wearable itself does not apply to most of the products we actually wear which are referred to as clothes. These are typically made of fabrics and have the a priori advantage of being in physical contact with large surface areas of the human body and already are a fact of life for all segments of society. As such, they present a significant opportunity[2,3] to harvest, store, and perhaps, even analyse relevant untapped physiologic variables. While uniquely positioned to address this challenge, the ability to impart digital attributes to fabrics has been limited.

To enable sensory, memory, and other digital functions[4–6] while retaining the traditional qualities that make fabric ubiquitous, one needs to consider intrinsic approaches for imparting digital functions to fabric constructions. Fibres, being the basic building blocks of fabrics, seem to be a natural candidate when compared with other approaches[7,8]. In recent years, a number of fibres with sophisticated functions have emerged, leading to sensing of various modalities[9–13], optical communication[14,15], actuators[16], and more[17,18]. However, these fibres without exception are analogue and lack digital componentry. In addition, up till now each functional fibre operated as single parallel tandem device. Under this constraint, the only way to achieve multiple functions was through multiple fibres which increases the required number of electrical access points, exposing the system to environmental and mechanical reliability challenges.

This study aims to introduce a fibre strand with a number of distinctive characteristics: the first is the introduction of digital components into a flexible polymeric fibre strand. The second is to lift the single-fibre, single-device limitation to allow a single fibre to deliver a scalable multiplicity of distinct addressable digital functions. The third is to enable access to the device ensemble internal to the fibre through a single connection port at the fibre's termination. The fourth is to enable sensory input collected by the fibre to be stored in the fibre itself. Last, we aim to store in the fibre not only sensory data but a neural network trained to infer context from it.

## Results

**Fabrication of digital fibres**. Drawn fibres contain continuous domains, presenting the opportunity to create uniform conductive buses connected to devices embedded along the entire length of the fibre. This allows for a reduction in the number of discrete electrical connections, which are a major source of mechanical and electrical vulnerability. Here, hundreds of individually addressable digital devices are electrically connected in situ during the fibre drawing process (i.e., *not* after draw), with all devices accessible on the same in-fibre digital communication bus. To construct this fibre, hundreds of square silicon microscale digital chips, each with four corner-positioned contact pads, are first placed into slots within a polymeric preform. Each chip is placed at angle of 26.56° with respect to the fibre axis (Fig. 1a) and the slots in the preform are milled to the exact dimensions of the chips (Fig. 1b and Supplementary Fig. 1) (See Methods for more information). This specific angle ensures the individual connection of four coplanar and coaxial tungsten wires to the four individual power and signal ports on each chip. It is critical

for this angle to be preserved under an angular window between 24.05° to 28.24° during the thermal draw to ensure consistent contact between the four axial wires and the individual pads (Fig. 1c) (See Supplementary Fig. 2 for the derivation of this critical angle range). A conventional limitation to the thermal-drawing approach, which complicates our fabrication, is that viscous extensional flow tends to align discrete particulates with the longitudinal axis[19] of the fibre as to lower their hydrodynamic cross section, while decreasing the viscosity of the fluid to reduce shear can lead to surface fluid instabilities resulting in random perturbations of the device positions. To achieve the conservation of the rotation angle, a viscous flow process window is identified (draw tension at 50–100 grams/mm$^2$) which preserves the angular orientation of the solid particulate components with high positional (microscale) and angular (±~2°) accuracy throughout the draw-down process.

During thermal drawing, four 25-μm diameter tungsten wires are *fed* into the preform. Compared to previous works[9–11] on thermal drawing of solder metal or conductive composites, this approach in utilising tungsten wires leads to higher axial conductivity along the fibre length. Importantly, the preform is deliberately designed so that as the features from preform to fibre are reduced in dimension during the draw process, these wires come closer together until they contact the underlying devices. We note that for the four wires to establish electrical contact with the silicon chips, the wires have to penetrate through an insulating polymeric barrier between the wires and devices present in the preform (Fig. 1b). To address this problem, the barrier between wires and devices (Fig. 1b, c) is chosen to be a softer polymer, polymethyl methacrylate (PMMA), which exhibits a lower viscosity at the draw temperature. During the preform-to-fibre transition, this combination of a soft polymer between hard polymer layers causes the harder outer polycarbonate (PC) cladding to compress the four electrode wires through the softer PMMA barrier until the wires are in electrical contact with the device pads. An additional wire (50 μm) is also fed under the device backing as support for levelling the devices along the fibre length, while adding a compressional force directed upwards from the bottom of device toward the four wires. This wire provides mechanical support for the device in the fibre, has no electrical contact with the device, and remains in the fibre after the draw. This dual approach of a soft-hard polymer combination and the backing wire enables robust wires-to-device electrical contacts. After drawing, the fibre contains an interconnected sequence of rotated devices, each with four un-shorted electrical connections to four electrode wires, across fibre lengths reaching tens of metres (Fig. 1d).

The extent of process control is indicated in the inset of Fig. 1d, which shows an array of fibres with internal devices all rotated at the ideal angle of 26.56°, with four 25 μm wires making electrical contact with four separate device pads (diameter ~100 μm) without shorting between wires. Supplementary Fig. 3 shows an array of fibres with embedded digital devices that are smaller in width. As illustrated in the size comparison with a coin in the Fig. 1d inset, the devices and connections are entirely packaged within the protective polymeric cladding, with the majority of the fibre body having a width of ~0.3 mm. The distance between the devices in the fibre ($d_{fibre}$) is dictated by two factors: the spacing between the devices in the preform ($d_{spacing}$) and the draw ratio (r), which is determined by the fraction of the preform diameter over the fibre diameter. In particular, $d_{fibre}$ is equal to $d_{spacing}$ multiplied by the square of r. In Supplementary Fig. 4, we show that devices can be spaced closely in the fibre with a distance of ~6.5 cm, by using a preform spacing of 0.65 mm and a draw ratio of 10. Further increase in device density can be achieved with a smaller device spacing in the preform and a lower draw down

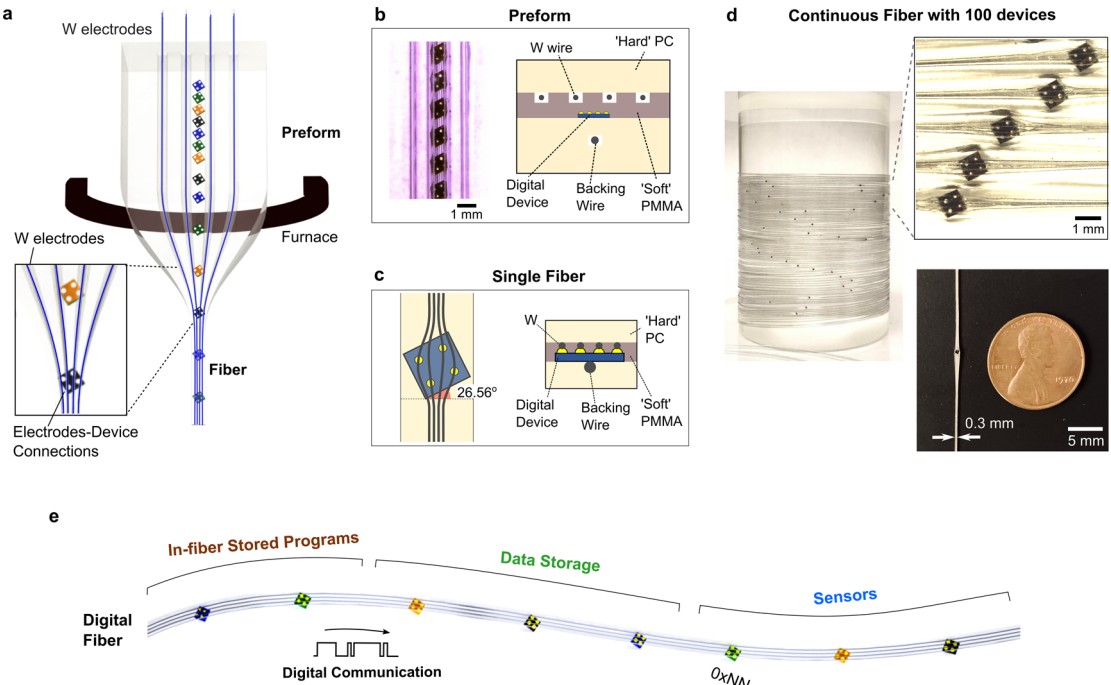

**Fig. 1 Fabrication approach of digital fibres. a** Thermal drawing of the digital fibre preform by feeding conductive tungsten wires into the empty channels. At the preform level, the tungsten (W) wires are spaced far apart with a polymeric barrier separating the device and the wires. The inset shows the converging of the four tungsten (W) electrodes toward the four electrical device pads at the necking region of the preform-to-fibre transition. **b** (left) Optical micrograph of the rotated devices embedded within preform slots milled at a critical angle of 26.56° relative to the preform axis. (right) Cross-sectional schematic of preform, containing three layers (hard PC/soft PMMA/hard PC). The presence of the hard PC enables an inwards compression that allows W electrodes to break through the soft PMMA barrier and make contact with the device electrical pads. The backing wire is used to compress the device on the opposite side from the electrode wires and to prevent tilting of the device along the fibre length. **c** (left) Schematic of drawn fibre with the rotation of embedded devices preserved at the critical angle of 26.56°, allowing for the electrical pads to contact four parallel wires without shorting. Spreading of the tungsten wires at the device region is also taken into account for the wires to align with these electrical pads. (right) Cross section schematic of the digital fibre at the device region, showing electrical contacts between device and wires. **d** (left) Photograph of a spool containing a continuous digital fibre with 100 embedded devices. (right) Magnified optical image of the fibre array showing that the digital devices are all rotated to the critical angle with connections to wire electrodes. (bottom) Photograph showing size comparison between the fibre and a coin (penny). **e** Illustration of the thermally drawn digital fibre encapsulating chips of different functionalities such as memory or sensing. Each of the chips is represented by a unique and different digital address in hexadecimal format (0xNN). The four wires within the fibre serve different functions: signal line for exchanging data between in-fibre devices, clock line to regulate communications signal timing, ground line, and power line.

ratio. A magnified view of the wire electrodes connected to the rotated device is shown in Supplementary Fig. 5a, and Supplementary Fig. 5b shows the statistics of rotational angles of devices within the fibre. The average recorded angle is 26.4° with a standard deviation as low as 0.9° across 157 devices, validating the scalable and reproducible control over each component's position and orientation. Within the range of angles for successful connections between 24.05° and 28.24°, we observe a 95.5% yield of discrete devices (i.e., 150 devices) with electrical pads that are properly aligned with the electrical wires within the fibre. In addition, improving the preform design from a fully polycarbonate (PC) version to a sandwiched preform (PC-PMMA-PC) increases the proportion of successfully connected devices (*n* = 50) from 60–90%, due to the effect of the soft-hard polymer combination. The remaining 10% remain unconnected due to the presence of a thin insulating polymer barrier. We note that this rotational approach for device interconnections also allows for devices with various pad configurations and different numbers of pads to be interconnected using different corresponding angles. For example, in Supplementary Fig. 6, we describe the connection of chips with a three-by-two pad configuration.

**Digital addressing and memory storage in fibres and fabrics.** Discrete in-fibre electronic devices are positioned uniformly at

different spatial positions along the fibre, with each device offering different functions such as sensing, data storage, or storage of in-fibre algorithms (Fig. 1e). By sending a unique serial combination of digital 1 and 0, the device with the correct matching digital address along the fibre can be switched 'on' to activate its internal functionality, which includes memory or sensing modalities. For this purpose, we implemented a digital addressing protocol, specifically the I²C protocol, within the fibre. The equivalent logic circuit of each device within the fibre is composed of XNOR and AND gates, as highlighted in Fig. 2a. Through this digital protocol, we show that multiple in-fibre devices can share the same four tungsten wire electrodes, yet by having distinct digital addresses, they are logically isolated from each other, hence we are able to independently address, access, and control separate devices and their corresponding functions at varying spatial positions along the fibre (shown at 5 cm, 40 cm, 70 cm) (Fig. 2a).

We note that these digital fibres are also thin and flexible, allowing them to be passed into a needle (Fig. 2b, i) and sewn into textiles (Fig. 2b, ii and iii). To illustrate the memory storage properties of these digital fibres and fabrics, Fig. 3a shows the writing, storing, and reading of binary information, to and from a fibre containing embedded digital memory devices, via digital signals communicated through its electrodes. For the use of these digital memory fibres in flexible fabric applications, we

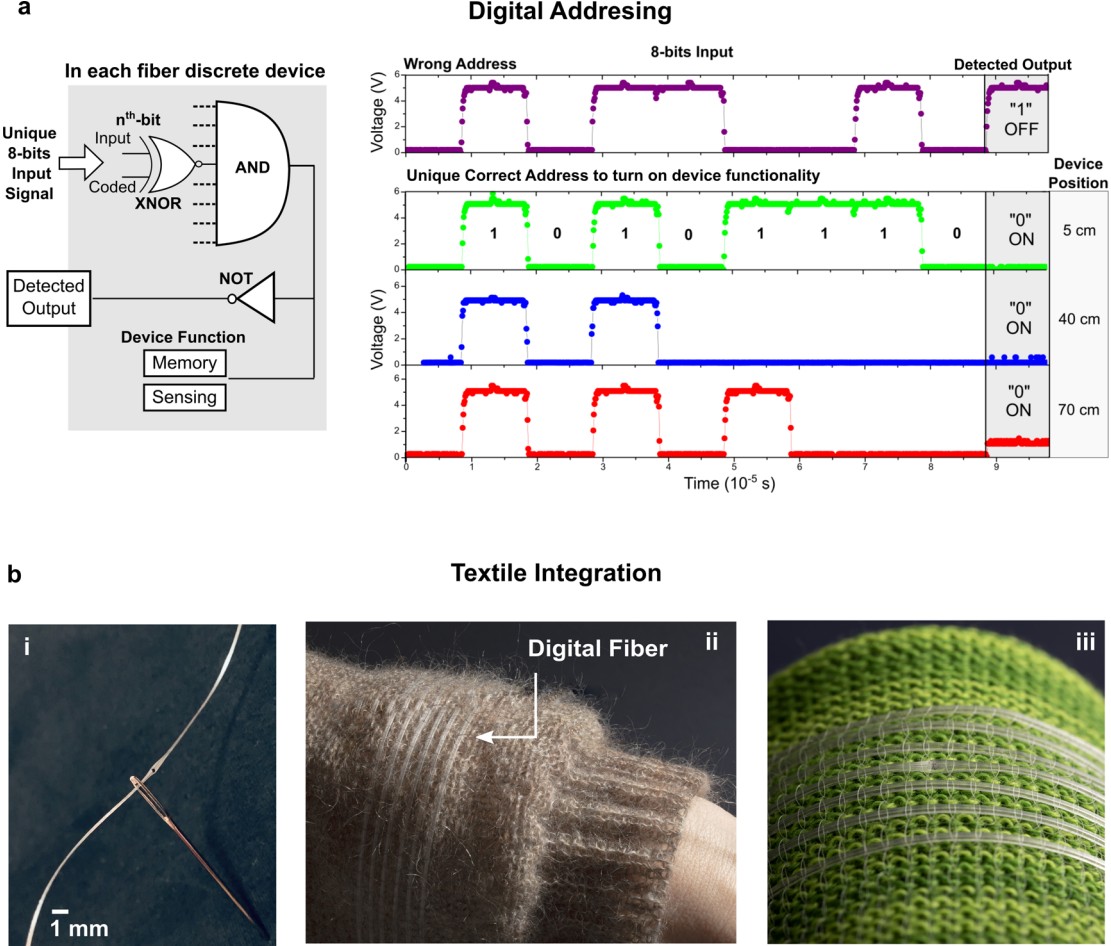

**Fig. 2 Digital addressing in fibre and textile integration. a** Individual digital addressing of different devices within the fibre. (Left) The digital circuit within each in-fibre device which enables the device to be activated if the input 8-bit signal matches its internal coded 8-bit identifier address. Each bit of the 8-bit input is read by a XNOR gate (Each dotted line represents a XNOR gate). The XNOR then compares a single input bit to the stored coded bit. If it matches, the output is 1. If all XNOR-outputs to the AND gate is one, the AND-output is then a one, activating the device function and pulling the signal line down via the NOT gate. (Right) Voltage-time plots of the digital fibre hosting multiple digital devices. Each of the devices is turned 'on' if the input matches its identifier address. Otherwise, the device remains 'OFF'. Note that each device is pulled up by the power line, hence the 'OFF' state is a logic one, and the 'ON' state is a logic 0. Each chip can be individually and separately controlled using its distinct digital address, allowing for spatial control of different device functions at different points along the fibre, for instance at 5 cm, 40 cm, and 70 cm. **b** Integration of the digital fibre (i) through a needle, (ii) in the sleeve of a sweater, and (iii) in a cotton-based fabric.

characterised their mechanical properties. First, the practicality of these memory fibres in real-life fabric applications is corroborated by their ability to withstand ten washing cycles with negligible degradation to voltage sharpness and zero loss of information stored in memory (Fig. 3b). Second, mechanical bending of the fibre to a curvature radius of 12 mm retains the sharp on-off voltage switching performance with zero loss of information in stored memory (Fig. 3c), highlighting the robustness of the fibre interconnects. To determine the radius of curvature at which either the wires break or start losing connections from the device, we bend a device-containing fibre segment around rods of decreasing radius, while measuring the resistance between two wire electrodes. When the tungsten wires are unbroken and well-connected to the device, the measured resistance is ~350 kiloohms. If the wires either break or lose connection to the device, the measured resistance will indicates an open circuit. We then bend the fibre around rods with progressively smaller radii and determine that the bending radius at which the fibre device becomes inoperable is 3.12 mm (Supplementary Fig. 7). Upon closer investigation through optical microscopy, it is found that the tungsten wires break and disconnect at this bending radius.

The digital memory fibre provides several significant advancements compared to previous studies based on analogue signals, including high switching performance, lower power requirements, and higher memory density. First, switching the digital state from zero to one requires only 3.3 V (Supplementary Fig. 8), with the power consumption during writing and reading new information measured to be 5.5 mW, as compared to 475 mW in previous analogue phase-change-memory chalcogenide fibres[20]. Second, the long-length scalability of thermal drawing, combined with the ability to individually operate numerous in-fibre devices, permits for large memory storage in a single strand of fibre. Supplementary Fig. 9 illustrates a 767-kilobit full-colour (red-green-blue) 8-frame movie file that is stored within a metre of fibre. This memory density value is approximately six orders of magnitude larger than the single-bit memory storage limit in previous thermally drawn fibres[20], arising from their single-fibre single-device constraint. By splitting information amongst the different addressable memory devices along the fibre (Fig. 3d), we then demonstrate storage of a music file (0.48 MBytes) within textiles (Fig. 3e). This advantage of a large memory capacity is further enhanced by the capability of the fibre to support

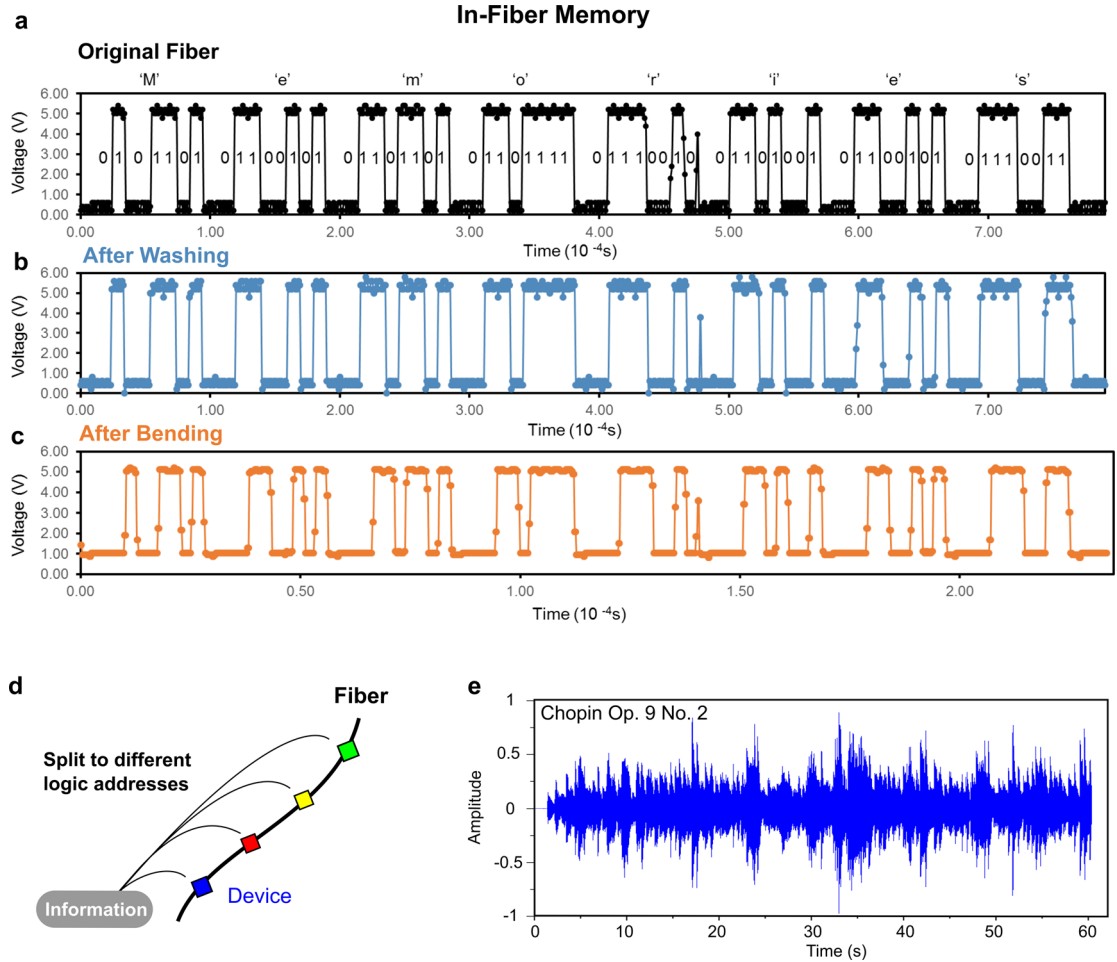

**Fig. 3 Storage of information in fibres. a** Writing and reading of binary information (the word 'Memories'), in and out of the fibre via digital signals communicated through its signal electrode, for the originally thermal drawn fibre (clock frequency:100 kHz), **b** reading of the fibre after washing for 10 cycles (clock frequency: 100 kHz), and **c** reading of the fibre after bending it with a curvature of radius of 12 mm (clock frequency: 300 kHz). **d** To store large information into the fibre, the information is split and stored into different memory devices along the fibre. Within 1 metre of fibre, 767 kilobits of information can be stored. **e** The plot of a 0.48 megabyte music piece written into the textile, stored for 2 months with no power, and later read out from the textile integrated with the memory fibres.

nonvolatile long-term data storage, as the movie and music were successfully stored in the fibre and fabric for 2 months without power.

**Physiological monitoring**. Another possibility afforded by this architecture involves *intra*-fibre communications between devices via digital signals. A digital fibre, composed of a hybrid of memory and temperature-sensing (MT) functionalities along the same strand, was fabricated. Figure 4a shows a schematic of the MT fibre with discrete digital thermistor devices interconnected with other memory units along the length of the fibre. In this fibre, temperature input is detected and converted from analogue to digital signals locally by the thermistor device. These digital signals are later communicated to the memory units within the fibre for storage. Figure 4b shows a 16-bit temperature reading which is sensed by the thermistor device and later stored within and read from the memory unit. We also note that this work demonstrates the first fibre strand which contains interconnected disparate functionalities that are addressable at different lengths of the fibre, unlike previous fibres[9–11] that are constrained to the same functionality along their length.

The combination of both memory and sensing functions presents opportunities for physiological applications. Specifically,

we demonstrate the use of this MT digital fibre, incorporated internally to the underside of a shirt to facilitate contact with the human skin, to store an individual's physiological data for health monitoring (Fig. 4c). Here, the fibre temperature sensor is in direct contact with the skin of the armpit. Body temperature is measured every 0.5 s and communicated to the fibre memory to store the temperature profile of distinct physical activities: sitting, standing, walking, and running. In the plot for indoor calibration, the initial rise in temperature (~1 °C increase) is attributed to the transient equalisation of the detected temperature by the sensor upon direct contact with the skin (armpit). After some time, a thermal equilibrium state is achieved hence resulting in a constant steady-state temperature. This transient increase in temperature is also reported in previous literature[21,22], where the rate of increase is related to how fast heat transfer can occur between the skin and the sensor. As evident in these prior literature and in our work, the time duration for this transient increase is typically in the range of minutes before steady-state temperature is recorded. Upon standing, the resting heart rate increases, causing a slight increase in body temperature, and the temperature–equalisation gradient occurs again. Next, the entire outdoor dataset, over 4.5 h of temperature measurements across multiple days, is detected in real-time and stored in the fibre digital memory itself, highlighting the long-term data-logging

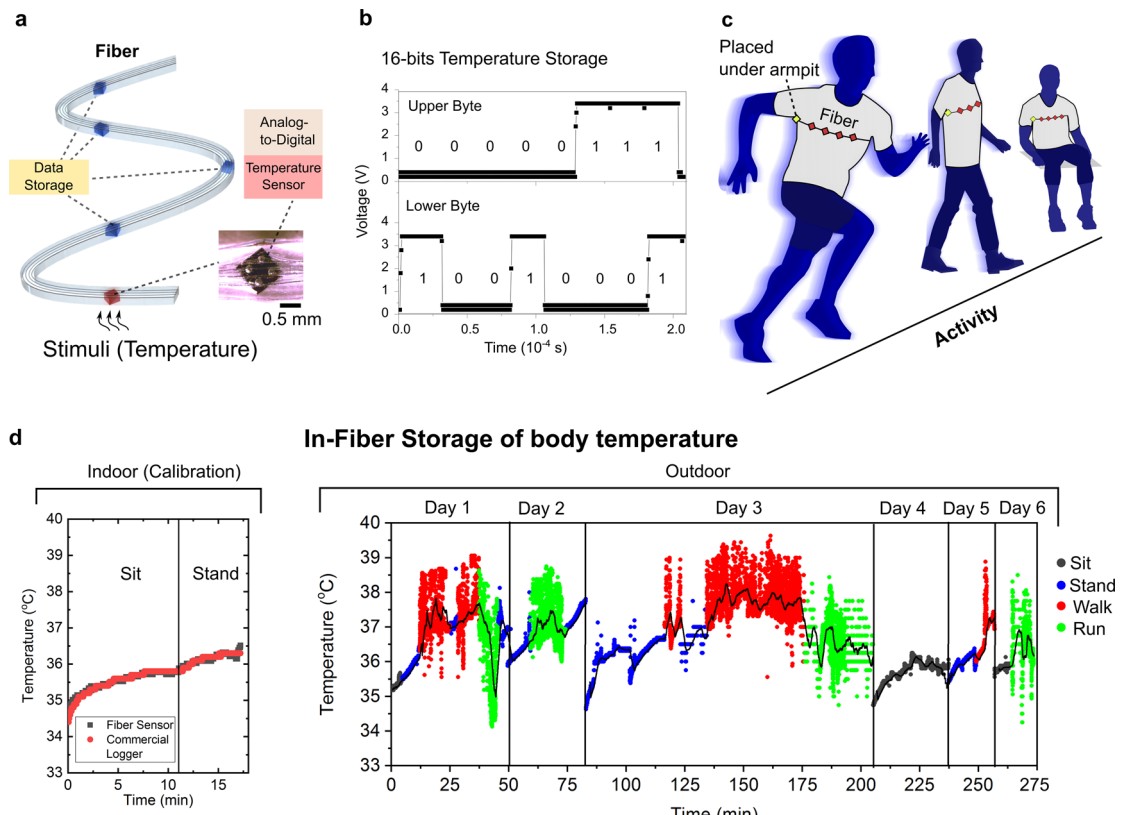

**Fig. 4 Hybrid digital fibres for physiological monitoring. a** Schematic of a hybrid digital memory and temperature-sensing (MT) fibre, containing a single thermistor device with multiple interconnected memory units along the fibre length. Upon temperature sensing, the analogue signal is converted to digital locally by the thermistor. The digital signals are processed by an external processor and transmitted to the in-fibre memory units for data storage. **b** Voltage–time plot of a 16-bit temperature measurement segmented into an upper and lower byte. The reading shown corresponds to a value of 1937, which is then converted to the decimal form of 19.37 during data analysis. **c** Schematic showing implementation of the MT fibre into a compression shirt that is worn by the user for storage of body temperature across different activities. Note that the thermistor unit is placed under the armpit to measure axillary body temperature. **d** Time plots of the in-fabric stored values of the user's body temperature. (left) The MT fabric temperature readings were calibrated with and validated against a commercial temperature logger and shown to correspond to reference measurements. (right) In-fabric storage of body temperature across 4.5 h (split up into multiple days) and across different activities. Uniquely, the MT fabric allows for outdoor activities, unlike conventional, bulky temperature-measurement equipment. Moreover, the MT fibre is flexible, permitting temperature recording across dynamic activities such as walking and running. The black line is the moving average over a 2-minute window. The lower temperature recorded during running is attributed to sweat cooling of the body.

capacity and mechanical robustness of the fibre (Fig. 4d). The oscillations in temperature during walking and running occur due to alternating heating and cooling while the arms are swinging. When the arm is in motion, the friction of motion in the armpit causes an increase in local environment temperature. Evaporative cooling of sweat at the armpit, which happens especially during running, result in a decrease in local temperature detected.

**Machine-learning inference in fabrics**. This body-temperature dataset is then used to train a neural network to detect and classify the signatures of sensory inputs, in this case the temperature–time patterns (Supplementary Fig. 10) of different human activities, enabling the classification of four distinct activities (sitting, standing, walking, and running) based on body temperature variations. To train this network, ~1800 sections of temperature values, each spanning 12 s corresponding to the four classes, are provided as input into a convolutional neural network. The number of layers and nodes and the values of weights and biases are optimised to provide high training accuracy (average of 97.9 ± 0.7% for 20 training processes) (see Methods for more details on the neural network). After training, the values of the weights and biases are extracted and reduced to two

significant figures to produce a compressed neural network. This compressed neural network, including mathematical equations for feature selection, weights, biases, and ReLu functions used in two convolutional filters, codes for matrix multiplication and addition used in layer operations, and weights, biases, and activation functions for the four-layer neural network (6 by 50 by 25 by 4 neurons, totalling 1650 neuronal connections), are all stored into the digital memory of the fibre within the fabric (Fig. 5a). Figure 5b shows the specific process flow of the neural network occurring in the fibre, from the input of sensed temperature data to the decision output. A 12-s temperature measurement sample corresponding to an unknown activity, initially sensed by the in-fibre thermistor, is provided as input to the compressed neural network within the digital fibre. The algorithms stored within the fibre memory then perform the following operations on the sample: first, selected features (standard deviation, averaged positive difference, and averaged negative difference) are calculated from the 12-s measurement. Next, these three input features are fed into two sequential layers of convolution with six filters, before performing flattening and max pooling. The outputs are then fed into the neural network with four layers to produce prediction probabilities of four different activities. The activity with highest probability is selected as the final output. These

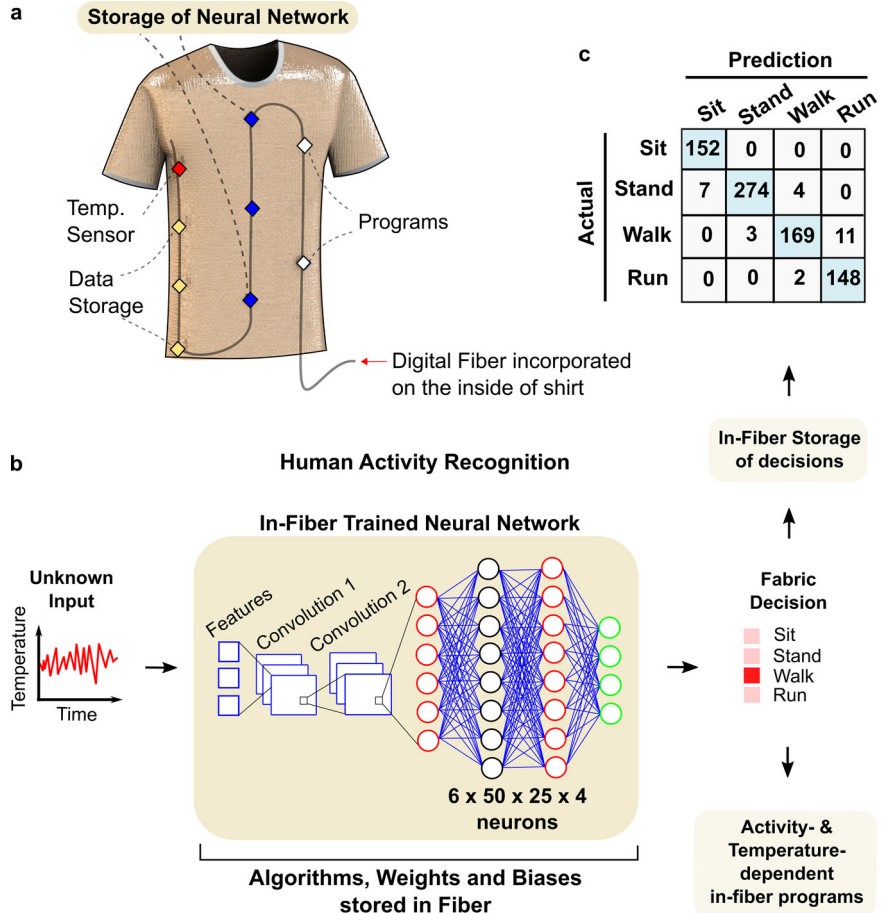

**Fig. 5 Fabric with incorporated neural network capabilities. a** Schematic of the shirt integrated with a digital fibre that has sensors, data storage, customisable programmes, and a neural network stored within its memory devices. **b** This digital fabric is capable of running a neural network that recognises what the wearer is doing in real-time without any human intervention. The operation of the neural network is as follows: a 12-s temperature measurement is measured in real-time and inputted into the in-fibre convolutional neural network (CNN) which is comprised of feature extraction, two convolutional layers, and four neuronal layers. After these layers, the fibre gives a final decision on the type of activity performed by the wearer. Upon decision-making, custom activity- and temperature-dependent programmes, stored in the fibre, provide feedback on the wearer's physiological status (Supplementary Fig. 11). **c** The accuracy of human activity recognition by the in-fabric CNN is shown by the prediction table, indicating an average accuracy of 96.4%.

decision outputs are then stored in the fibre. To evaluate the accuracy of the decision-making, the output decision is compared against the actual activity. The in-fabric neural network for human activity recognition achieves test accuracy of 96.4% (Fig. 5c). We illustrate a number of potential applications of fabrics with machine-learning inference capabilities: For instance, real-time activity tracking for individuals, coupled with the ability to detect temperature abnormalities, may provide early warning signs for conditions such as hyperthermia (Supplementary Fig. 11) or assist in early detection of illnesses such as COVID-19 in larger populations.

## Discussion

In summary, tens of metres of polymeric fibre exhibiting digital sensing and digital memory are introduced with an overall storage capacity of $\sim 7.6 \times 10^5$ memory bits per metre. A single connection at the end of the fibre is used to access multiple devices independently, enabling the operation of multiple functions from a single fibre, hence overcoming the single-fibre single-device limitation faced by prior approaches. Reducing the number of required electrical connections to a single terminal addresses a major source of environmental and reliability vulnerability. The utility of this approach is demonstrated by using a single fibre to collect and store 270 min of surface body temperature data across multiple days and during a variety of physical activities. A neural network with 1650 neuronal connections is trained on the basis of the temperature-time-activity correlation and is subsequently stored in the same fibre. When presented with an unknown temperature-time data set, the fibre infers the type of physical activity with an accuracy of 96% based on a 12-s temperature reading. The ability to measure and store digital information in a fibre which also contains inference algorithms presents intriguing opportunities in a variety of applications in which functional fibres and fabrics are in close proximity with the human body, including autonomous drug delivery[23], neural interfaces[18], personal thermal management[24], and computing in fabrics[2].

## Methods

**Preform preparation.** Digital microchip-embedded fibres are drawn from a macroscopic rectangular preform. The thickness, width, and length of the preform are ~11, ~12, and ~200 mm respectively. We place memory and temperature sensor microchips in predefined micro-pockets at the critical angle condition and utilise the soft–hard sandwiching design to achieve a firm connection of pads to wires during the thermal drawing process.

*Material selection.* We choose transparent thermoplastic polycarbonate (PC) due to its excellent mechanical properties for fibre and fabric device applications, and

further, its role of top and bottom sandwiching layer during the thermal fibre drawing. As the soft middle layer, we utilise Poly(methyl methacrylate) (PMMA). PMMA has a lower viscosity than PC at the fibre drawing temperature which makes it a suitable sandwiched soft polymer necessary for the wire electrodes to cut through easily during the draw.

*Programmable micromachining.* We use a CNC milling machine and programme a desired number of microscale pockets, using an irregular pocket function which provides the critical angle orientation for the microchips. A PMMA slab is first cut in dimensions of 12 mm width, 0.8 mm thickness, and 100 mm length, on which micro pockets are defined on the front face of the slab. Pocket dimensions are 0.84 mm × 0.84 mm for temperature sensors, 0.50 mm × 0.50 mm for lower-capacity memory devices, and 0.84 mm × 0.90 mm for larger-capacity memory devices. Each device pocket is milled to a depth of 0.4 mm. The distance between micro-chips is programmed to be 1.15 mm, which after drawing results in an inter-device spacing of approximately 5 cm to 20 cm depending on the draw down ratio. For this study, preform were created with chip counts ranging from 50 to 150 in a single fibre. On the back face of the PMMA slab, we mill four channels with dimensions of 0.25 mm × 0.25 mm × 200 mm to enable the feeding of 25-μm tungsten wires. The centre to centre distance between channels is 0.84 mm. This number is critical for achieving simultaneous and accurate connection between wires and pads without electrical shorting or position mismatching. We discover this value by considering the draw-down ratio from preform to fibre and considering the flow behaviour of wires around the devices during the preform to fibre transition. We utilise a 0.5 mm endmill for rough and fast milling and 0.125 mm endmill for fine milling to ensure that the microchips firmly sit in the pocket at the correct angle. We design the functional layer to be exactly in the centre of the sandwiched preform design to facilitate uniform force applied by the PC layers. For the preform with the bigger memory devices, the top sandwiching PC layer (bare slab) has a thickness of 4.8 mm, a width of 12 mm and a length of 200 mm and approaches the PMMA layer from the pad side of the microchips. The bottom sandwiching PC layer (thickness of 5.2 mm and length of 200 mm) contains a channel of 0.52 mm × 0.52 mm × 200 mm dimensions. This channel is used to feed 50 μm tungsten as a backing wire which aids in pressing the microchip onto four contact wires during the preform to fibre transition. To prevent direct contact of the back wire to the microchips, we use a 0.25 mm thick PC layer (width of 12 mm and length of 200 mm) between the PMMA and bottom PC slab. We also use two 50 mm long PC layers (with the same width and thickness of the functional PMMA layer) and place them at the top and bottom tips of the PMMA layer to prevent deformation of the functional layer during the preform consolidation process. For a thinner fibre with the smaller memory devices, the width and thickness of the slabs used above is scaled smaller by a factor of four, with the final assembled preform cross sectional dimension being approximately 4 mm × 3 mm.

*Preform consolidation.* We assembled the milled components and inserted sacrificial Teflon coated wires into the channels on PMMA and PC layers. This ensures that the channels are preserved in shape during the consolidation process. We consolidated the preform at 186 °C for 55 min in the oven to bond the PC and PMMA layers together.

**Thermal drawing of digital fibres with rotated devices.** The preform is drawn in a three-zone vertical tube furnace with a top-zone temperature of 110°, a middle-zone temperature of 248 °C (bait-off temperature is 270 °C), and a bottom-zone temperature of 110 °C while feeding four 25 μm electrical connection wires and one 50-μm tungsten back-wire. The fibre dimensions are monitored with laser micrometres. We use a stress of 50–100 grams/mm$^2$ to draw the fibre. The draw-down ratio of the process is designed to be 10 to 15. The preform feed rate is 1 mm/min and capstan speed varies depending on the draw down ratio. The fibre cross-sectional size is 300 μm by 225 μm. The fibre tower system is capable of scalable manufacturing of these digital fibres with a current fibre production speed of 10 m/min.

**Electrical characterisation of digital fibres.** The main digital memory chips used are 24CW1280X integrated circuits from Microchip Technology. We have also validated fibres with smaller-capacity memory chips such as the AT24CSW01. We wrote custom code to change the digital address of the memory chips. To make connections to the end of the digital fibre, we dissolve the fibre end in dichloromethane to remove the cladding. We then remove the backing wire and separate the four wire electrodes to make individual connections to external readout circuitry. To determine if the chips are connected, we wrote scanner code that finds chips of different addresses along the fibre. We then use an oscilloscope to register and validate the digital waveforms on the signal and clock lines as we transmit input signals and receive acknowledgement signals from the in-fibre chips.

**Storage of movie and music in the memory fibres.** The movie is first split into eight frames. We then use a Python programme to extract the Red-Green-Blue (RGB) intensity values (0 to 255) for each pixel in the frames. These values are presented in a list format, separated by spaces. We make use of a 'writing' programme to encode these integer values into the memory chips in the fibre. Each value takes up one byte, which is stored at a single memory address within the 8-bit

memory chip. The memory architecture is that of an electrically erasable programmable read-only memory (EEPROM). To validate the non-volatility of the fibre-embedded EEPROM modules, this information is stored in the fibre, without power, for 2 months. A memory 'reading' programme is then utilised to read the information stored in all of the memory addresses in the fibre. The final read information is in a form of a list of numbers, which is fed back into another Python programme which reconstructs this list back into the individual RGB frames. For the music file, the music is first pre-processed into 16-bit values at a sample rate of 4000 Hz. Similarly, a writing and reading programme for 16-bit values is used to write, store and read the individual audio samples into and from the memory fibre.

**Fabrication and operation of hybrid digital memory-temperature (MT) fibres.** The digital temperature-sensing chips are MAX31875 thermistors from Maxim Integrated. These MT fibres are drawn using the same thermal drawing parameters as the memory fibres. To test if all devices are connected, we run the scanner code. We then write a programme that configures and operates the thermistor chips to sense a new temperature value every 0.5 s. The collected temperature-sensing values are continuously written into the in-fibre memory chips for storage of the body temperature during operation. This programme is written in JavaScript and is stored in the in-fibre memory chips in a string format. To run this programme, the programme is interpreted by an external microprocessor (width ~1 cm) (STM32F401). An external LiPo battery is used to power the MT Fibres at 3.3 V. The MT fibres are manually sewn into the compression shirt and held onto the compression shirt through tight woven bridges. To ensure that that the digital fibre is in continuous contact with the skin, the tight compression shirt is used to compress the fibre sensor toward the torso. In addition, it is designed for the fibre temperature sensor to be situated at the uppermost point of the armpit because the sensor can then be situated between the upper arm and the torso, enabling compression of the sensor by the arms toward the torso at all times even during walking and running. The flexibility of the polymer fibre allows it to be integrated with the compression shirt and facilitates conformability of the digital fibres against the skin during movement. The commercial temperature logger used for calibration and validation is an AZ Instruments Corp 88598.

**Bending and washing conditions.** To test resilience of the fibres to bending, the digital memory fibre is bent around a stainless steel rod with a diameter of 24 mm. The stored information is then read from the digital memory of the fibre to determine if the wires are still connected to the device and if data is still retained in the memory device. To determine the radius of curvature at which the wires lose connection to the device, we perform a destructive test by bending the fibre around stainless steel rods with progressively smaller diameters, measuring the resistance across the fibre electrodes with an ohmmeter to determine the point at which physical failure causes a disconnection from the embedded devices.

For the washing test, the memory fibres were placed in a laundry bag and washed with a portable washing machine (Pyle PUCWM11). The memory fibres were subjected to 10 washing cycles ("cotton" wash mode, no detergent, 15 min duration per cycle) at a water temperature of 45 °C.

**Neural network and programmes in the digital fibres.** To train the neural network, the stored data of the user's body temperature was first extracted from the MT digital fibres. This data was then labelled according to the different activities. We wrote a data processing programme that segments the stored data into sections of desired durations. The stored data was split into consecutive sample sections of 12-s duration, with an overlap of 6-s between each sections. Each 12-s section contains 24 data points as the temperature is logged every 0.5 s. For each of these sections, we selected and calculated three features: the standard deviation (STD), the average positive difference (PCX), and the average negative difference (NCX). The average difference is first measured by taking the difference between the $n$ and $n + 1$ temperature values. If it is positive, it goes into a 'positive' list from which the list average is calculated to be PCX. If it is negative, it goes into a 'negative' list from which the list average is calculated to be NCX. To train the neural network, we utilise 1794 sections labelled with different activities. The training and initial validation was done offline via Keras on an external computer (Python in PyCharm). We trained an one-dimensional convolutional neural network (CNN) with two convolution layers, max pooling, flattening, and four neuronal layers. The inputs to the CNN are the three features (STD, PCX, NCX). The 1794 sections are split into 90% training data and 10% validation data. The number of epochs used was 1000, stopping when there were signs of overfitting, i.e., when the training loss curve diverged away from the validation loss curve. Both convolution layers used a filter size of six with a kernel size of one. The activation type is ReLu. For max pooling, the pool size is 2. For the neural network, the first layer uses six neurons, the second layer uses 50 neurons, the third layer uses 25, and the fourth layer uses four neurons. All four layers utilise ReLu as the activation function. The four outputs for the different activities (sitting, standing, walking, and running) are then converted into probabilities by the softmax function. The neural network was optimised using the categorical cross-entropy loss function and Adam Optimisation Algorithm. The above parameters were optimised to fit the model within the storage capacity of the in-fibre memory and reduce decision-making latency while preventing accuracy loss from over-compression. In addition, the three features

(STD, PCX, NCX) are used, instead of other statistical measurements, because they give the highest accuracy with small neuronal network size. After training, the trained weights and biases are compressed to two significant figures to reduce memory usage. This compressed neural network, including mathematical equations for features selection, weights, biases, and ReLu functions used in two convolutional filters, codes for matrix multiplication and addition, and weights and biases for the $6 \times 50 \times 25 \times 4$ neural network layers, were all stored into fibre. The prediction accuracy was obtained by inputting the test data through the compressed neural network within the fibre and comparing its output with the actual activity. The number of test sections is 770, comprised of different activities. All programmes for the different physiological applications, as well as the voice commands, as demonstrated in Supplementary Fig. 10, are stored in and read out from the memory fibre as a string format.

## Data availability

The data that support the findings of this study are available from the corresponding author upon reasonable request.

## Code availability

Source codes of the programmes and algorithms used for this study are available from the corresponding author upon reasonable request.

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

## Acknowledgements

This work was supported by the MIT MRSEC through the MRSEC Programme of the National Science Foundation under award number DMR-1419807, the US Army Research Laboratory and the US Army Research Office through the Institute for Soldier Nanotechnologies, under contract number W911NF-13-D-000, the MIT Sea Grant under the contract number NA18OAR4170105, and the Defence Threat Reduction Agency through the Department of Defence under the contract number SA21-03. The authors acknowledge Ayelet Karmon, CIRTex Centre for Textile Research, and Shenkar College for providing access to a knitting facility and Roni Cnaani for photographing the fibre-integrated textiles. The authors acknowledge Samuel Fuller for his advice on signalling and performance of digital devices in fibres. The authors also acknowledge Grace Noel for her assistance in proof-reading the article.

## Author contributions

G.L. and Y.F. conceived of the idea of applying digital fibres toward artificially intelligent fabrics. G.L. and T.K. designed and implemented the experiments toward the drawing of the digital fibres. G.L., J.F., W.Y. and T.K. contributed to the electrical and material characterisation of the fibres. B.W., S.F., I. Chinn, and P.-W.C. contributed to the training of the AI programme and implementation and storage of the AI code in the memory fibres. S.F., I. Chatziveroglou, and S.P. contributed to the writing of the temperature-sensing programmes. Y.S. and I. Chin contributed to the writing of the programme that writes, stores, and reads music files into and from the fibres. J.J. contributed to the ideas that optimize the thermal drawing experiments and applications of these digital fibres. A.G.-K. conceived, designed, and machine-knitted the textiles integrating the fibres.

## Competing interests

The authors declare no competing interests.
