## [Peer Review File · Nature Communications]

Reviewers' Comments:

Reviewer #1:

Remarks to the Author:

This work integrates microelectronic chips (sensor, memory) inside a fiber via preform-thermal-drawing fiber making route, and demonstrates the wearable functions of collecting temperature and store it in memory chips for subsequent processing. The memory units work well after ten washing cycles and bending to a curvature radius of 12 mm. The fabrics made of such fibers have been used for garments to collect on-body data continuously with a high-accuracy machine-learning inference of human activity through a deep neural network stored within the fiber. Comparing with flexible organic electronics, they are more reliable and more accurate. The present paper contains detailed information on its fabrication and testing.

Thus I agree with the authors that the digital fibers may open new opportunities in the fields of fiberelectronics, personal computing, and intelligent textiles.

Reviewer #2:

Remarks to the Author:

The authors demonstrate a scalable fabrication approach to incorporate in-situ digital functionalities into thin flexible polymeric fibers. The precise control on both positions and angles of digital chips enables the formation of in-fiber digital electronics over the entire length of fiber and further large-scale fabrics, achieving physiological sensing, data storage, and more importantly on-body data analytics with machine-learning inference. The reported results are novel and interesting to the field, and the paper is very well organized and presented. I recommend that this paper can be accepted after addressing the following few comments:

1. Figure 1d shows a spool containing a continuous digital fiber with 100 embedded devices. What are the factors to determine the device density (the number of devices can be hosted in a meter long fiber)? What are the potential solutions to further increase device density?
2. The dual approach using a soft-hard polymer combination and the backing wire is brilliant to enable wires-to-device contacts. A 95.5% (150/157) yield rate for successful electrical connection was achieved. The authors stated that the failed connections were due to the rotation of the chip angle. I think the failed connections might be also caused by the situation that the insulating barrier between the wires and devices can not be completely compressed away, even softer polymer was selected to construct such a layer.
3. In addition to temperature sensing, what other physiological signals can be measured?

Reviewer #3:

Remarks to the Author:

This article reports an innovative and scalable approach to incorporate hundreds of microscale electronics devices into polymeric fiber strands with length up to tens of meters. The authors harness the positions and angles of these functional devices during fiber drawing process, which shows more than 95% incorporating yield. After knitting into a shirt, it can be used for physiological sensing, data storage and AI analysis. Overall, the quality of the work and the manuscript are very high, and will undoubtedly be of interest to a broad audience. I would recommend it to be published after minor revision.

1. Since the process is very promising, I recommend adding supporting videos to show details of the fabrication process.
2. in the experimental part, please include the details of washing conditions and bending conditions.
3. To show the robustness of these digital fibers in harsh conditions, please provide 1-2 destructive experiments that most devices would disconnect from W wires.
4. When these digital fibers are worn on a body and attached to the skin for temperature sensing

and data storage (Shown in Figure 4d), the temperature shows large variation during movement (walk and run). Can the authors explain in details the reason? How about the conformability of these digital fibers to the skin during movement? The left plot of Figure 4d shows the temperatures increasing continuously when in sit and stand features. Can the authors explain it?

*Reviewers' comments are in **bold**, the authors' responses are in Roman. The pertinent (modified) text within the manuscript is indicated in blue.*

Reviewer #1 (Remarks to the Author):

This work integrates microelectronic chips (sensor, memory) inside a fiber via preform-thermal-drawing fiber making route, and demonstrates the wearable functions of collecting temperature and store it in memory chips for subsequent processing. The memory units work well after ten washing cycles and bending to a curvature radius of 12 mm. The fabrics made of such fibers have been used for garments to collect on-body data continuously with a high-accuracy machine-learning inference of human activity through a deep neural network stored within the fiber. Comparing with flexible organic electronics, they are more reliable and more accurate. The present paper contains detailed information on its fabrication and testing.

Thus I agree with the authors that the digital fibers may open new opportunities in the fields of fiber electronics, personal computing, and intelligent textiles.

Response: We thank the reviewer for the positive comments. We agree about the promising reliability and accuracy of fiber electronics in garments, and we are excited about this work opening new opportunities in personal computing and intelligent textiles.

Reviewer #2 (Remarks to the Author):

The authors demonstrate a scalable fabrication approach to incorporate in-situ digital functionalities into thin flexible polymeric fibers. The precise control on both positions and angles of digital chips enables the formation of in-fiber digital electronics over the entire length of fiber and further large-scale fabrics, achieving physiological sensing, data storage, and more importantly on-body data analytics with machine-learning inference. The reported results are novel and interesting to the field, and the paper is very well organized and presented. I recommend that this paper can be accepted after addressing the following few comments:

Response: We thank the reviewer for the positive comments in regards to the novelty and importance of this work to the field of fibers and fabrics, as well as the positive remarks on the organization and presentation of this paper.

1. Figure 1d shows a spool containing a continuous digital fiber with 100 embedded devices. What are the factors to determine the device density (the number of devices can be hosted in a meter long fiber)? What are the potential solutions to further increase device density?

Response: We thank the reviewer for this comment on the device density. The distance between the devices in the fiber (d_{fiber}) is dictated by two factors: the spacing between the devices in the preform (d_{spacing}) and the draw down ratio (r), which is determined by the fraction of the preform diameter over the fiber diameter. In particular, d_{fiber} is equal to the multiplication of d_{spacing} and the square of r . In Supplementary Figure 4, we show that devices can be spaced in

fiber with a distance of only 6.5 cm, by using a preform spacing of 0.65 mm and a draw ratio of 10, hence achieving 15 devices in a meter-long fiber. To further increase device density, devices in the preform can be spaced closer together and by using a lower draw-down ratio.

We have include the following text to highlight the device density in the main text under the section titled “Fabrication of Digital Fibers”:

“The distance between the devices in the fiber (d_{fiber}) is dictated by two factors: the spacing between the devices in the preform ($d_{spacing}$) and the draw ratio (r), which is determined by the fraction of the preform diameter over the fiber diameter. In particular, d_{fiber} is equal to $d_{spacing}$ multiplied by the square of r . In Supplementary Figure 4, we show that devices can be spaced in the fiber with a distance of ~6.5 cm, by using a preform spacing of 0.65 and a draw ratio of 10. Further increase in device density can be achieved with a smaller device spacing in the preform and a lower draw down ratio.”

2. The dual approach using a soft-hard polymer combination and the backing wire is brilliant to enable wires-to-device contacts. A 95.5% (150/157) yield rate for successful electrical connection was achieved. The authors stated that the failed connections were due to the rotation of the chip angle. I think the failed connections might be also caused by the situation that the insulating barrier between the wires and devices can not be completely compressed away, even softer polymer was selected to construct such a layer.

Response: We thank the reviewer for this comment on the insulating barrier. We agree that in certain cases, the insulating barrier is not be completely compressed away. From our experiments, we have determined that improving the preform, from a fully polycarbonate (PC) version to a sandwiched preform (PC-PMMA-PC), increases the number of connected devices ($n = 50$) from 60% to 90%, in which the remaining 10% is not connected due to the presence of a thin insulating barrier. In regards to the percentage of 95.5%, this value is pertaining to the number of devices that has electrical pads correctly rotated with respect to the position of the electrical wires within the fiber. To further clarify this distinction, we have revised the previous sentence on 95.5% in the main text as well as added more information on the yield of connected devices, as follows:

In the text, section titled “Fabrication of Digital Fibers”:

“Within the range of angles for successful connections between 24.05° and 28.24° , we observe a 95.5% yield of discrete devices (i.e. 150 devices) with electrical pads that are properly aligned with the electrical wires within the fiber. In addition, improving the preform design from a fully polycarbonate (PC) version to a sandwiched preform (PC-PMMA-PC), increases the proportion of successfully connected devices ($n = 50$) from 60% to 90%, due to the effect of the soft-hard polymer combination. The remaining 10% remain unconnected due to the presence of a thin insulating polymer barrier.”

3. In addition to temperature sensing, what other physiological signals can be measured?

Response: We thank the reviewer for this comment on other forms of physiological signals. As part of a future work, discrete digital accelerometer chips can also be integrated into the fiber, which allow for the collection of vibration and movement physiological signals, such as heartbeat and body gait measurements.

Reviewer #3 (Remarks to the Author):

This article reports an innovative and scalable approach to incorporate hundreds of microscale electronics devices into polymeric fiber strands with length up to tens of meters. The authors harness the positions and angles of these functional devices during fiber drawing process, which shows more than 95% incorporating yield. After knitting into a shirt, it can be used for physiological sensing, data storage and AI analysis. Overall, the quality of the work and the manuscript are very high, and will undoubtedly be of interest to a broad audience. I would recommend it to be published after minor revision.

Response: We thank the reviewer for the positive comments on the high quality of this work as well as the potential of this work undoubtedly being of interest to a broad audience.

1. Since the process is very promising, I recommend adding supporting videos to show details of the fabrication process.

Response: We thank the reviewer for the affirmation of the potential of the fabrication process. We have added a supporting video (Supplementary Video 1) to give more insights to the fabrication process.

2. in the experimental part, please include the details of washing conditions and bending conditions.

Response: We thank the reviewer for this comment on including the details of the washing and bending conditions. We have included the following text in the Methods section:

“Bending and washing condition

To test resilience of the fibers to bending, the digital memory fiber is bent around a stainless steel rod with a diameter of 24 mm. The stored information is then read from the digital memory of the fiber to determine if the wires are still connected to the device and if data is still retained in the memory device. To determine the radius of curvature at which the wires lose connection to the device, we perform a destructive test by bending the fiber around stainless steel rods of progressively smaller diameters, measuring the resistance across the fiber electrodes with an ohmmeter to determine the point at which physical failure causes a disconnection from the embedded devices.

For the washing test, the memory fibers were placed in a laundry bag and washed with a portable washing machine (Pyle PUCWM11). The memory fibers were subjected to 10 washing cycles (“cotton” wash mode, no detergent, 15 minute duration per cycle) at temperature of 45 °C.

“

3. To show the robustness of these digital fibers in harsh conditions, please provide 1-2 destructive experiments that most devices would disconnect from W wires.

Response: We thank the reviewer for this comment on showing the robustness of these digital fibers in harsh conditions. We have performed a destructive experiment where the fiber with devices is bent with much smaller radius of curvature until the wires lose connection from the

device. We have found that the devices only disconnect from the wires when the fibers are very harshly bent, with a radius of curvature of only 3.12 mm, indicating the robustness of the fiber. We have included this information in the main text, as well as described the experiment in the Method section.

In the main text, under Digital Addressing and Memory Storage in Fibers and Fabrics:

“Second, mechanical bending of the fiber to a curvature radius of 12 mm retains the sharp on-off voltage switching performance with zero loss of information in stored memory (Fig. 3c), highlighting the robustness of the fiber interconnects. To determine the radius of curvature at which either the wires break or start losing connections from the device, we bend a device-containing fiber segment around rods of decreasing radius, while measuring the resistance between two wire electrodes. When the tungsten wires are unbroken and well-connected to the device, the measured resistance is ~350 kilohms. If the wires either break or lose connection to the device, the measured resistance indicates an open circuit. We then bend the fiber around rods with progressively smaller radii and determine that the bending radius, at which the fiber device becomes inoperable, is 3.12 mm (Supplementary Fig. 7). Upon closer investigation through optical microscopy, it is found that the tungsten wires break and disconnect at this bending radius.”

In the Method section under Bending and Washing Condition:

“To test the resilience of the fibers to bending, the digital memory fiber is bent around a stainless steel rod with a diameter of 24 mm. The stored information is then read from the digital memory of the fiber to determine if the wires are still connected to the device and if data is still retained in the memory device. To determine the radius of curvature at which the wires lose connection to the device, we perform a destructive test by bending the fiber around stainless steel rods with progressively smaller diameters, measuring the resistance across the fiber electrodes with an ohmmeter to determine the point at which physical failure causes a disconnection from the embedded devices.”

4. When these digital fibers are worn on a body and attached to the skin for temperature sensing and data storage (Shown in Figure 4d), the temperature shows large variation during movement (walk and run). Can the authors explain in details the reason? How about the conformability of these digital fibers to the skin during movement? The left plot of Figure 4d shows the temperatures increasing continuously when in sit and stand features. Can the authors explain it?

Response: We thank the reviewer for this comment on temperature variations during dynamic movements, conformability of the fibers to the skin, and the continuous increase in temperature during sitting and standing. The large oscillations in temperature during walking and running occurs when the arms are swinging, due to the alternating effects of (1) local increase in temperature caused by friction, and the (2) cooling from sweat evaporation through the armpit when the arm is swinging during movement. To further clarify the paper, we have included the following text in the main text to describe the sources of these variations:

In the main text, under the section “Physiological Monitoring”:

“The oscillations in temperature during walking and running occur due to alternating heating and cooling while the arms are swinging. When the arm is in motion, the friction of motion in the armpit causes an increase in the local environment temperature. Evaporative cooling of sweat at the armpit, which happens especially during running, results in a decrease in local temperature detected.”

To ensure that the digital fiber is in continuous contact with the skin, we have made use of a tight compression shirt to hold the fiber sensor against the body. In addition, it is designed for the fiber temperature sensor to be situated at the uppermost point of the armpit because the sensor can then be situated between the upper arm and the torso (armpit), enabling compression of the sensor by the arms towards the torso at all times even during walking and running. The flexibility of the polymer fiber allows it to be integrated with the compression shirt and facilitates conformability of the fibers against the skin during movement. To further clarify the paper, we have included the following text in the Methods section to describe the implementation of the fiber sensor:

In the Methods section under “Fabrication and Operation of Hybrid Digital Memory-Temperature (MT) Fibers”

”To ensure that the digital fiber is in continuous contact with the skin, the tight compression shirt is used to compress the fiber sensor towards the torso. In addition, it is designed for the fiber temperature sensor to be situated at the uppermost point of the armpit because the sensor can then be situated between the upper arm and the torso, enabling compression of the sensor by the arms towards the torso at all times even during walking and running. The flexibility of the polymer fiber allows it to be integrated with the compression shirt and facilitates conformability of the digital fibers against the skin during movement.”

Regarding the rise in temperature observed using both the fiber sensor and the commercial temperature logger during the sitting and standing calibration, the initial rise in temperature is attributed to the transient increase of the detected temperature by the sensor upon direct contact with the skin (armpit). After some time, a thermal equilibrium state is achieved hence resulting in a constant steady-state temperature towards the end of the “sit” section. It is also important to note that this increase during equalization is actually only a small increase in measured temperature (~ 1 °C). This transient increase in temperature is also reported in previous literature^{i,iii}, where the rate of increase is related to how fast heat transfer can occur between the skin and the sensor. As evident in these mentioned literature and in our work (as validated by both the fiber sensor and the commercial logger), the time duration for this transient increase is typically in the range of minutes before steady-state temperature is recorded. Upon standing, the resting heart rate increases, causing a slight increase in body temperature, and the temperature-equalization gradient again occurs. We have included the following text in the main text in order to further elaborate these initial transient increase in temperature:

In the main text, under the section “Physiological Monitoring”

“In the plot for indoor calibration, the initial rise in temperature (~ 1 °C increase) detected by the fiber and commercial logger is attributed to the transient equalization of the detected temperature by the sensor upon direct contact with the skin (armpit). After some time, a thermal equilibrium state is achieved hence resulting in a constant steady-state temperature. This transient increase in temperature is also reported in previous literatures, where the rate of increase is related to how fast heat transfer can occur between the skin and the sensor. As evident in these mentioned literature and in our work, the time duration for this transient increase is typically in the range of minutes before steady-state temperature is recorded. Upon standing, the resting heart rate increases, causing a slight increase in body temperature, and the temperature-equalization gradient occurs again.”

ⁱ Smith, A. D. H., Crabtree, D. R., Bilzon, J. L. J. & Walsh, N. P. The validity of wireless iButtons® and thermistors for human skin temperature measurement. *Physiol. Meas.* (2010). doi:10.1088/0967-3334/31/1/007

ⁱⁱ Liu, H. *et al.* The response of human thermal perception and skin temperature to step-change transient thermal environments. *Build. Environ.* (2014). doi:10.1016/j.buildenv.2013.12.007

Reviewers' Comments:

Reviewer #2:

Remarks to the Author:

All my comments have been well addressed. This submission is ready for publication in Nature Communications.

Reviewer #3:

Remarks to the Author:

The authors have revised the manuscript taking into accounts all my major points. It reads nicely. I support publication in the current form.

REVIEWERS' COMMENTS

Reviewer #2 (Remarks to the Author):

All my comments have been well addressed. This submission is ready for publication in Nature Communications.

Reviewer #3 (Remarks to the Author):

The authors have revised the manuscript taking into accounts all my major points. It reads nicely. I support publication in the current form.

Authors' Response: We thank the reviewers for dedicating their time and effort in reviewing this work and for their positive comments and appreciation. Their comprehensive review has helped to improve the flow and scientific merits of this publication.